# Scale-invariant Learning by Physics Inversion

**Philipp Holl** [*]
Technical University of Munich

**Vladlen Koltun**
Apple

**Nils Thuerey**
Technical University of Munich

## Abstract

Solving inverse problems, such as parameter estimation and optimal control, is a vital part of science. Many experiments repeatedly collect data and rely on machine learning algorithms to quickly infer solutions to the associated inverse problems. We find that state-of-the-art training techniques are not well-suited to many problems that involve physical processes. The highly nonlinear behavior, common in physical processes, results in strongly varying gradients that lead first-order optimizers like SGD or Adam to compute suboptimal optimization directions. We propose a novel hybrid training approach that combines higher-order optimization methods with machine learning techniques. We take updates from a scale-invariant inverse problem solver and embed them into the gradient-descent-based learning pipeline, replacing the regular gradient of the physical process. We demonstrate the capabilities of our method on a variety of canonical physical systems, showing that it yields significant improvements on a wide range of optimization and learning problems.

## 1   Introduction

Inverse problems that involve physical systems play a central role in computational science. This class of problems includes parameter estimation [54] and optimal control [57]. Among others, solving inverse problems is integral in detecting gravitational waves [22], controlling plasma flows [38], searching for neutrinoless double-beta decay [3, 1], and testing general relativity [19, 35].

Decades of research in optimization have produced a wide range of iterative methods for solving inverse problems [47]. Higher-order methods such as limited-memory BFGS [36] have been especially successful. Such methods compute or approximate the Hessian of the optimization function in addition to the gradient, allowing them to locally invert the function and find stable optimization directions. Gradient descent, in contrast, only requires the first derivative but converges more slowly, especially in ill-conditioned settings [49].

Despite the success of iterative solvers, many of today's experiments rely on machine learning methods, and especially deep neural networks, to find unknown parameters given the observations [11, 15, 22, 3]. While learning methods typically cannot recover a solution up to machine precision, they have a number of advantages over iterative solvers. First, their computational cost for inferring a solution is usually much lower than with iterative methods. This is especially important in time-critical applications, such as the search for rare events in data sets comprising of billions of individual recordings for collider physics. Second, learning-based methods do not require an initial guess to solve a problem. With iterative solvers, a poor initial guess can prevent convergence to a global optimum or lead to divergence (see Appendix C.1). Third, learning-based solutions can be less prone to finding local optima than iterative methods because the parameters are shared across a large collection of problems [30]. Combined with the stochastic nature of the training process, this allows gradients from other problems to push a prediction out of the basin of attraction of a local optimum.

---

[*]Corresponding author, `philipp.holl@tum.de`

36th Conference on Neural Information Processing Systems (NeurIPS 2022).

Practically all state-of-the-art neural networks are trained using only first order information, mostly due to the computational cost of evaluating the Hessian w.r.t. the network parameters. Despite the many breakthroughs in the field of deep learning, the fundamental shortcomings of gradient descent persist and are especially pronounced when optimizing non-linear functions such as physical systems. In such situations, the gradient magnitudes often vary strongly from example to example and parameter to parameter.

In this paper, we show that inverse physics solvers can be embedded into the traditional deep learning pipeline, resulting in a hybrid training scheme that aims to combine the fast convergence of higher-order solvers with the low computational cost of backpropagation for network training. Instead of using the adjoint method to backpropagate through the physical process, we replace that gradient by the update computed from a higher-order solver which can encode the local nonlinear behavior of the physics. These physics updates are then passed on to a traditional neural network optimizer which computes the updates to the network weights using backpropagation. Thereby our approach maintains compatibility with acceleration schemes [18, 33] and stabilization techniques [32, 31, 7] developed for training deep learning models. The replacement of the physics gradient yields a simple mathematical formulation that lends itself to straightforward integration into existing machine learning frameworks.

In addition to a theoretical discussion, we perform an extensive empirical evaluation on a wide variety of inverse problems including the highly challenging Navier-Stokes equations. We find that using higher-order or domain-specific solvers can drastically improve convergence speed and solution quality compared to traditional training without requiring the evaluation of the Hessian w.r.t. the model parameters.

## 2    Scale-invariance in Optimization

We consider unconstrained inverse problems that involve a differentiable physical process $\mathcal{P} : X \subset \mathbb{R}^{d_x} \to Y \subset \mathbb{R}^{d_y}$ which can be simulated. Here $X$ denotes the physical parameter space and $Y$ the space of possible observed outcomes. Given an observed or desired output $y^* \in Y$, the inverse problem consists of finding optimal parameters

$$x^* = \arg\min_x L(x) \quad \text{with} \quad L(x) = \frac{1}{2}\|\mathcal{P}(x) - y^*\|_2^2. \tag{1}$$

Such problems are classically solved by starting with an initial guess $x_0$ and iteratively applying updates $x_{k+1} = x_k + \Delta x_k$. Newton's method [5] and many related methods [10, 36, 23, 41, 46, 8, 12, 6] approximate $L$ around $x_k$ as a parabola $\tilde{L}(x) = L(x_k) + \frac{\partial L(x_k)}{\partial x_k}(x - x_k) + \frac{1}{2}H_k(x - x_k)^2$ where $H_k$ denotes the Hessian or an approximation thereof. Inverting $\tilde{L}$ and walking towards its minimum with step size $\eta$ yields

$$\Delta x_k = -\eta \cdot H_k^{-1} \left(\frac{\partial L(x_k)}{\partial x_k}\right)^T. \tag{2}$$

The inversion of $H$ results in scale-invariant updates, i.e. when rescaling $x$ or any component of $x$, the optimization will behave the same way, leaving $L(x_k)$ unchanged. An important consequence of scale-invariance is that minima will be approached equally quickly in terms of $L$ no matter how wide or deep they are.

Newton-type methods have one major downside, however. The inversion depends on the Hessian $H$ which is expensive to compute exactly, and hard to approximate in typical machine learning settings with high-dimensional parameter spaces [25] and mini-batches [51].

Instead, practically all state-of-the-art deep learning relies on first-order information only. Setting $H$ to the identity in Eq. 2 yields gradient descent updates $\Delta x = -\eta \cdot \left(\frac{\partial L}{\partial x}\right)^T$ which are not scale-invariant. Rescaling $x$ by $\lambda$ also scales $\Delta x$ by $\lambda$, inducing a factor of $\lambda^2$ in the first-order-accurate loss change $L(x) - L(x + \Delta x) = -\eta \cdot \left(\frac{\partial L}{\partial x}\right)^2 + \mathcal{O}(\Delta x^2)$. Gradient descent prescribes small updates to parameters that require a large change to decrease $L$ and vice-versa, typically resulting in slower convergence than Newton updates [56]. This behavior is the root cause of exploding or vanishing gradients in deep neural networks. The step size $\eta$ alone cannot remedy this behavior whenever $\frac{\partial L}{\partial x}$

varies along $x$. Figure 1 shows the optimization trajectories for the simple problem $\mathcal{P}(x) = (x_1, x_2^2)$ to illustrate this problem.

When training a neural network, the effect of scaling-variance can be reduced through normalization in intermediate layers [32, 7] and regularization [37, 53] but this level of control is not present in most other optimization tasks, such as inverse problems (Eq. 1). More advanced first-order optimizers try to solve the scaling issue by approximating higher-order information [33, 29, 18, 39, 40, 55, 45, 50], such as Adam where $H \approx \operatorname{diag}\left(\left|\frac{\partial L}{\partial x}\right|\right)$, decreasing the loss scaling factor from $\lambda^2$ to $\lambda$. However, these methods lack the exact higher-order information which limits the accuracy of the resulting update steps when optimizing nonlinear functions.

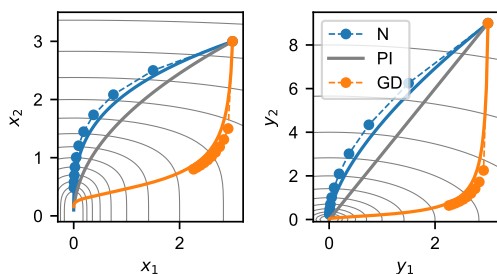

Figure 1: Minimization (Eq. 1) with $y \equiv \mathcal{P}(x) = (x_1, x_2^2)$. $L$ contours in gray. Trajectories of gradient descent (GD), Newton's method (N), and perfect physics inversion (PI) shown as lines (infinitesimal $\eta$) and circles (10 iterations with constant $\eta$).

## 3 Scale-invariant Physics and Deep Learning

We are interested in finding solutions to Eq. 1 using a neural network, $x^* = \mathrm{NN}(y^* \,|\, \theta)$, parameterized by $\theta$. Let $\mathcal{Y}^* = \{y_i^* \,|\, i = 1, ..., N\}$ denote a set of $N$ inverse problems involving $\mathcal{P}$. Then training the network means finding

$$\theta_* = \arg\min_\theta \sum_{i=1}^{N} \frac{1}{2} \|\mathcal{P}\big(\mathrm{NN}(y_i^* \,|\, \theta)\big) - y_i^*\|_2^2. \tag{3}$$

Assuming a large parameter space $\theta$ and the use of mini-batches, higher-order methods are difficult to apply to this problem, as described above. Additionally, the scale-variance issue of gradient descent is especially pronounced in this setting because only the network part of the joint problem $\mathrm{NN} \circ \mathcal{P}$ can be properly normalized while the physical process $\mathcal{P}$ is fixed. Therefore, the traditional approach of computing the gradient $\frac{\partial L}{\partial \theta}$ using the adjoint method (backpropagation) can lead to undesired behavior.

Consider the problem $\mathcal{P}(x) = e^x$ with observed data $y^* \in (0, 1]$ (Appendix C.2). Due to the exponential form of $\mathcal{P}$, the curvature around the solutions $x^* = \log(y^*)$ strongly depends on $y^*$. This causes first-order network optimizers such as SGD or Adam to fail in approximating the correct solutions for small $y^*$ because their gradients are overshadowed by larger $y^*$ (see Fig. 2). Scaling the gradients to unit length in $x$ drastically improves the prediction accuracy, which hints at a possible solution: If we could employ a scale-invariant physics solver, we would be able to optimize all examples, independent of the curvature around their respective minima.

### 3.1 Derivation

If $\mathcal{P}$ has a unique inverse and is sufficiently well-behaved, we can split the joint optimization problem (Eq. 3) into two stages. First, solve all inverse problems individually, constructing a set of unique solutions $\mathcal{X}^{\mathrm{sv}} = \{x_i^{\mathrm{sv}} = \mathcal{P}^{-1}(y_i^*)\}$ where $\mathcal{P}^{-1}$ denotes the inverse problem solver. Second, use $\mathcal{X}^{\mathrm{sv}}$ as labels for supervised training

$$\theta_* = \arg\min_\theta \sum_{i=1}^{N} \frac{1}{2} \|\mathrm{NN}(y_i^* \,|\, \theta) - x_i^{\mathrm{sv}}\|_2^2. \tag{4}$$

This enables scale-invariant physics (SIP) inversion while a fast first-order method can be used to train the network which can be constructed to be normalized using state-of-the-art procedures [32, 7, 53].

Unfortunately, this two-stage approach is not applicable in multimodal settings, where $x^{\mathrm{sv}}$ depends on the initial guess $x_0$ used in the first stage. This would cause the network to interpolate between possible solutions, leading to subpar convergence and generalization performance. To avoid these

problems, we alter the training procedure from Eq. 4 in two ways. First, we reintroduce $\mathcal{P}^{-1}$ into the training loop, yielding

$$\theta_* = \arg\min_\theta \sum_{i=1}^{N} \frac{1}{2}\|\mathrm{NN}(y_i^*\,|\,\theta) - \mathcal{P}^{-1}(y_i^*)\|_2^2\,. \tag{5}$$

Next, we condition $\mathcal{P}^{-1}$ on the neural network prediction by using it as an initial guess, $\mathcal{P}^{-1}(y^*) \to \mathcal{P}^{-1}(y^*\,|\,\mathrm{NN}(y^*\,|\,\theta))$. This makes training in multimodal settings possible because the embedded solver $\mathcal{P}^{-1}$ searches for minima close to the prediction of the NN. Therefore $\theta$ can exit the basin of attraction of other minima and does not need to interpolate between possible solutions. Also, since all inverse problems from $\mathcal{Y}^*$ are optimized jointly, this reduces the likelihood of any individual solution getting stuck in a local minimum, as discussed earlier.

The obvious downside to this approach is that $\mathcal{P}^{-1}$ must be run for each training step. When $\mathcal{P}^{-1}$ is an iterative solver, we can write it as $P^{-1}(y^*\,|\,\mathrm{NN}(y^*\,|\,\theta)) = \mathrm{NN}(y^*\,|\,\theta) + \Delta x_0 + ... + \Delta x_n$. We denote the first update $\Delta x_0 \equiv U(y^*\,|\,\theta)$.

Instead of computing all updates $\Delta x$, we approximate $P^{-1}$ with its first update $U$. Inserting this into Eq. 5 with $(\circ) \equiv (y_i^*\,|\,\theta)$ yields

$$\theta_* = \arg\min_\theta \sum_{i=1}^{N} \frac{1}{2}\|\mathrm{NN}(y_i^*\,|\,\theta) - (\mathrm{NN}(\circ) + U(\circ))\|_2^2\,. \tag{6}$$

This can be further simplified to $\sum_{i=1}^{N}\frac{1}{2}\|U(y_i^*\,|\,\theta)\|_2^2$ but this form is hard to optimize directly as is requires backpropagation through $U$. In addition, its minimization is not sufficient, because all fixed points of $U$, such as maxima or saddle points of $L$, can act as attractors.

Instead, we make the assumption $\frac{\partial \mathcal{P}^{-1}}{\partial y} = 0$ to remove the $(\circ)$ dependencies in Eq. 6, treating them as constant. This results in a simple $L^2$ loss for the network output.

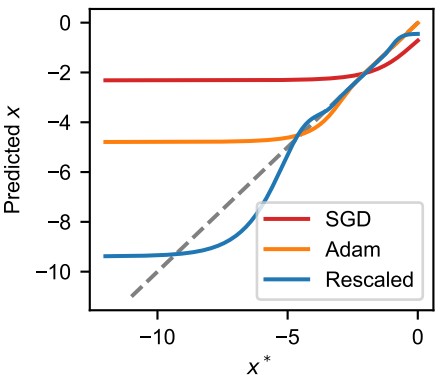

Figure 2: Networks trained according to Eq. 3 with $\mathcal{P}(x) = e^x$. Stochastic gradient descent (SGD) and Adam fail to approximate solutions for small values due to scale variance. Normalizing the gradient in $x$ space (Rescaled) improves solution accuracy by decreasing scale variance.

As we will show, this avoids both issues. It also allows us to factor the optimization into a network and a physics graph (see Fig. 3), so that all necessary derivative computations only require data from one of them.

## 3.2 Update Rule

The factorization described above results in the following update rule for training with SIP updates, shown in Fig. 3:

1. Pass the network prediction $x_0$ to the physics graph and compute $\Delta x_0 \equiv U(y_i^*\,|\,x_0)$ for all examples in the mini-batch.

2. Send $\tilde{x} \equiv x_0 + \Delta x_0$ back and compute the network loss $\tilde{L} = \frac{1}{2}\|x_0 - \tilde{x}\|_2^2$.

3. Update $\theta$ using any neural network optimizer, such as SGD or Adam, treating $\tilde{x}$ as a constant.

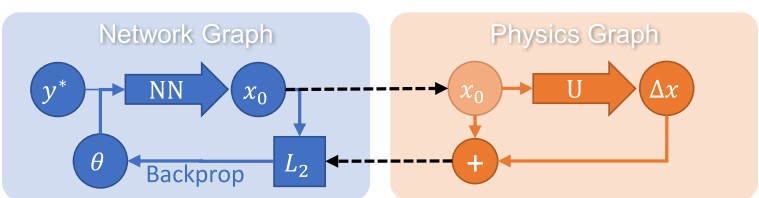

Figure 3: Neural network (NN) training procedure with embedded inverse physics solver (U).

To see what updates $\Delta\theta$ result from this algorithm, we compute the gradient w.r.t. $\theta$,

$$\frac{\partial \tilde{L}}{\partial \theta} = \sum_{i=1}^{N} U(y^* \,|\, x_0) \cdot \frac{\partial \mathrm{NN}}{\partial \theta} \tag{7}$$

Comparing this to the gradient resulting from optimizing the joint problem $\mathrm{NN} \circ \mathcal{P}$ with a single optimizer (Eq. 3),

$$\frac{\partial L}{\partial \theta} = \Delta y_i \cdot \frac{\partial \mathcal{P}}{\partial x} \frac{\partial \mathrm{NN}}{\partial \theta} \tag{8}$$

where $\Delta y_i = \mathcal{P}\big(\mathrm{NN}(y_i^* \,|\, \theta)\big) - y_i^*$, we see that $U$ takes the place of $\Delta y_i \cdot \frac{\partial \mathcal{P}}{\partial x}$, the adjoint vector that would otherwise be computed by backpropagation. Unlike the adjoint vector, however, $\tilde{x} = x_0 + U$ encodes an actual physical configuration. Since $\tilde{x}$ can stem from a higher-order solver, the resulting updates $\Delta\theta$ can also encode non-linear information about the physics without computing the Hessian w.r.t. $\theta$.

### 3.3 Convergence

It is a priori not clear whether SIP training will converge, given that $\frac{\partial \mathcal{P}^{-1}}{\partial y}$ is not computed. We start by proving convergence for two special choices of the inverse solver $\mathcal{P}^{-1}$ before considering the general case. We assume that NN is expressive enough to fit our problem and that it is able to converge to every point $x$ for all examples using gradient descent:

$$\exists \eta > 0 : \forall i \,\forall x \,\forall \epsilon > 0 \,\exists n \in \mathbb{N} : ||\mathrm{NN}_{\theta_n} - x||_2 \leq \epsilon \tag{9}$$

where $\theta_n$ is the sequence of gradient descent steps

$$\theta_{n+1} = \theta_n - \eta \left(\frac{\partial \mathrm{NN}}{\partial \theta}\right)^T (\mathrm{NN}_{\theta_n} - x) \tag{10}$$

with $\eta > 0$. Large enough networks fulfill this property under certain assumptions [17] and the universal approximation theorem guarantees that such a configuration exists [13].

In the first special case, $U(x) \equiv U(y^* \,|\, x)$ points directly towards a solution $x^*$. This models the case of a known ground truth solution in a unimodal setting. For brevity, we will drop the example indices and the dependencies on $y^*$.

**Theorem 1.** *If* $\forall x \,\exists \lambda \in (0, 1] \,:\, U(x) = \lambda(x^* - x)$, *then the gradient descent sequence* $\mathrm{NN}_{\theta_n}$ *with* $\theta_{n+1} = \theta_n + \eta \left(\frac{\partial \mathrm{NN}}{\partial \theta}\right)^T U(x)$ *converges to* $x^*$.

*Proof.* Rewriting $U(x) = -\frac{\partial}{\partial x}\left(\frac{\lambda}{2}||x - x^*||_2^2\right)$ yields the update $\theta_{n+1} - \theta_n = -\eta \left(\frac{\partial \hat{L}}{\partial x} \frac{\partial \mathrm{NN}}{\partial \theta}\right)^T$ where $\hat{L} = \frac{\lambda}{2}||x - x^*||_2^2$. This describes gradient descent towards $x^*$ with the gradients scaled by $\lambda$. Since $\lambda \in (0, 1]$, the convergence proof of gradient descent applies. $\square$

The second special case has $U(x)$ pointing in the direction of steepest gradient descent in $x$ space.

**Theorem 2.** *If* $\forall x \,\exists \lambda \in (0, 1] \,:\, U(x) = -\lambda \left(\frac{\partial L}{\partial x}\right)^T$, *then the gradient descent sequence* $\mathrm{NN}_{\theta_n}$ *with* $\theta_{n+1} = \theta_n + \eta \left(\frac{\partial \mathrm{NN}}{\partial \theta}\right)^T U(x)$ *converges to minima of* $L$.

*Proof.* This is equivalent to gradient descent in $L(\theta) \equiv (\mathrm{NN} \circ L)(\theta)$. Rewriting the update yields $\theta_{n+1} - \theta_n = -\eta\lambda \left(\frac{\partial L}{\partial x} \frac{\partial \mathrm{NN}}{\partial \theta}\right)^T$ which is the gradient descent update scaled by $\lambda$. $\square$

Next, we consider the general case of an arbitrary $\mathcal{P}^{-1}$ and $U$. We require that $U$ decreases $L$ by a minimum relative amount specified by $\tau$,

$$\exists \tau > 0 \,:\, \forall x \,:\, L(x) - L(x + U(x)) \geq \tau \left(L(x) - L(x^*)\right). \tag{11}$$

To guarantee convergence to a region, we also require

$$\exists K > 0 \,:\, \forall x \,:\, ||U(x)|| \leq K(L(x) - L(x^*)). \tag{12}$$

**Theorem 3.** *There exists an update strategy $\theta_{n+1} = S(\theta_n)$ based on a single evaluation of $U$ for which $L(\mathrm{NN}_{\theta_n}(y))$ converges to a minimum $x^*$ or minimum region of $L$ for all examples.*

*Proof.* We denote $\tilde{x}_n \equiv x_n + U(x_n)$ and $\Delta L^* = L(x_n) - L(x^*)$. Let $I_2$ denote the open set of all $x$ for which $L(x) - L(x_n) > \frac{\tau}{2}\Delta L^*$. Eq. 11 provides that $\tilde{x}_n \in I_2$. Since $I_2$ is open, $\exists \epsilon > 0 : \forall x \in B_\epsilon(\tilde{x}_n) : L(x_n) - L(x) > \frac{\tau}{2}\Delta L^*$ where $B_\epsilon(x)$ denotes the open set containing all $x' \in \mathbb{R}^d$ for which $||x' - x||_2 < \epsilon$, i.e. there exists a small ball around $\tilde{x}_n$ which is fully contained in $I_2$ (see sketch in Fig. 4).

Using Eq. 9, we can find a finite $n \in \mathbb{N}$ for which $\mathrm{NN}_{\theta_n} \in B_\epsilon(\tilde{x}_n)$ and therefore $L(x_n) - L(\mathrm{NN}_{\theta_n}) > \frac{\tau}{2}\Delta L^*$. We can thus use the following strategy $S$ for minimizing $L$: First, compute $\tilde{x}_n = x_n + U(x_n)$. Then perform gradient descent steps in $\theta$ with the effective objective function $\frac{1}{2}||\mathrm{NN}_\theta - \tilde{x}_n||_2^2$ until $L(x_n) - L(\mathrm{NN}_\theta) \geq \frac{\tau}{2}\Delta L^*$. Each application of $S$ reduces the loss to $\Delta L_{n+1}^* \leq (1 - \frac{\tau}{2})\Delta L_n^*$ so any value of $L > L(x^*)$ can be reached within a finite number of steps. Eq. 12 ensures that $||U(x)|| \to 0$ as the optimization progresses which guarantees that the optimization converges to a minimum region. $\square$

While this theorem guarantees convergence, it requires potentially many gradient descent steps in $\theta$ for each physics update $U$. This can be advantageous in special circumstances, e.g. when $U$ is more expensive to compute than an update in $\theta$, or when $\theta$ is far away from a solution. However, in many cases, we want to re-evaluate $U$ after each update to $\theta$. Without additional assumptions about $U$ and $\mathrm{NN}_\theta$, there is no guarantee that $L$ will decrease every iteration, even for infinitesimally small step sizes. Despite this, there is good reason to assume that the optimization will decrease $L$ over time. This can be seen visually in Fig. 4 where the next prediction $x_{n+1}$ is guaranteed to lie within the blue circle. The region of increasing loss is shaded orange and always fills less than half of the volume of the circle, assuming we choose a sufficiently small step size. We formalize this argument in appendix A.2.

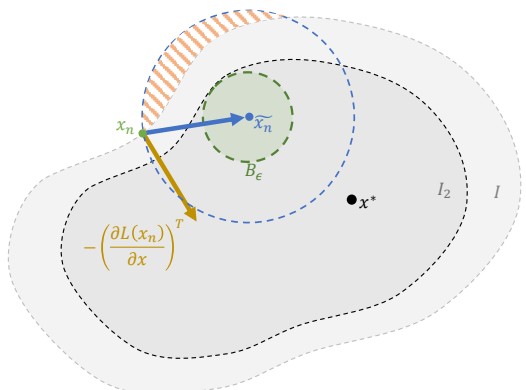

Figure 4: Convergence visualization for the proof of theorem 3. All shown objects are visualized in $x$ space for one example $i$.

**Remarks and lemmas** We considered the case that $U \equiv \Delta x_0$. This can be trivially extended to the case that $U \equiv \Delta x_0 + ... + \Delta x_m$ for any $m \in \mathbb{N}$. When we let the solver run to convergence, i.e. $m$ large enough, theorem 1 guarantees convergence if $\mathcal{P}^{-1}$ consistently converges to the same $x_i^*$. Also note that any minimum $\theta^*$ found with SIP training fulfills $U = 0$ for all examples, i.e. we are implicitly minimizing $\sum_{i=1}^{N} \frac{1}{2}||U(y_i^* \,|\, \theta)||_2^2$.

### 3.4 Experimental Characterization

We first investigate the convergence behavior of SIP training depending on characteristics of $\mathcal{P}$. We construct the synthetic two-dimensional inverse process

$$\mathcal{P}(x) = (\sin(\hat{x}_1)/\xi, \ \hat{x}_2 \cdot \xi) \quad \text{with} \quad \hat{x} = \gamma \cdot R_\phi \cdot x,$$

where $R_\phi \in \mathrm{SO}(2)$ denotes a rotation matrix and $\gamma > 0$. The parameters $\xi$ and $\phi$ allow us to continuously change the characteristics of the system. The value of $\xi$ determines the conditioning of $\mathcal{P}$ with large $\xi$ representing ill-conditioned problems while $\phi$ describes the coupling of $x_1$ and $x_2$. When $\phi = 0$, the off-diagonal elements of the Hessian vanish and the problem factors into two independent problems. Fig. 5a shows one example loss landscape.

We train a fully-connected neural network to invert this problem (Eq. 3), comparing SIP training using a saddle-free Newton solver [14] to various state-of-the-art network optimizers. We select the best learning rate for each optimizer independently. For $\xi = 0$, when the problem is perfectly conditioned, all network optimizers converge, with Adam converging the quickest (Fig. 5b). Note

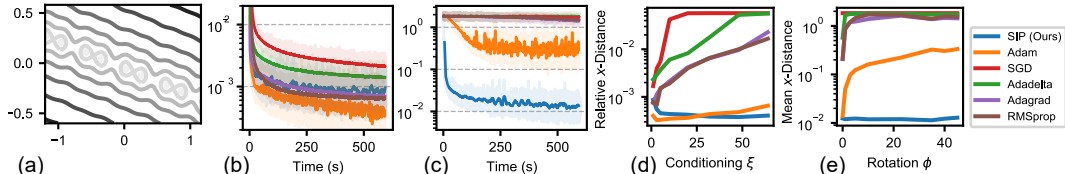

Figure 5: (a) Example loss landscape with $y^* = (0.3, -0.5)$, $\xi = 1$, $\phi = 15°$. (b,c) Learning curves with $\phi = \frac{\pi}{4}$, averaged over 256 min-batches. For (b) $\xi = 1$, and (c) $\xi = 32$. (d) Dependence on problem conditioning $\xi$ (with $\phi = 0$). (e) Dependence on parameter coupling $\phi$ (with $\xi = 32$).

that the relatively slow convergence of SIP mostly stems from it taking significantly more time per iteration than the other methods, on average 3 times as long as Adam. As we have spent little time in eliminating computational overheads, SIP performance could likely be significantly improved to near-Adam performance.

When increasing $\xi$ with $\phi = 0$ fixed (Fig. 5d), the accuracy of all traditional network optimizers decreases because the gradients scale with $(1/\xi, \xi)$ in $x$, elongating in $x_2$, the direction that requires more precise values. SIP training uses the Hessian to invert the scaling behavior, producing updates that align with the flat direction in $x$ to avoid this issue. This allows SIP training to retain its relative accuracy over a wide range of $\xi$. At $\xi = 32$, only SIP and Adam succeed in optimizing the network to a significant degree (Fig. 5c).

Varying $\phi$ with $\xi = 32$ fixed (Fig. 5e) sheds light on how Adam manages to learn in ill-conditioned settings. Its diagonal approximation of the Hessian reduces the scaling effect when $x_1$ and $x_2$ lie on different scales, but when the parameters are coupled, the lack of off-diagonal terms prevents this. SIP training has no problem with coupled parameters since its updates are based on the full-rank Hessian $\frac{\partial^2 L}{\partial x^2}$.

### 3.5 Application to High-dimensional Problems

Explicitly evaluating the Hessian is not feasible for high-dimensional problems. However, scale-invariant updates can still be computed, e.g. by inverting the gradient or via domain knowledge. We test SIP training on three high-dimensional physical systems described by partial differential equations: Poisson's equation, the heat equation, and the Navier-Stokes equations. This selection covers ubiquitous physical processes with diffusion, transport, and strongly non-local effects, featuring both explicit and implicit solvers. All code required to reproduce our results is available at `https://github.com/tum-pbs/SIP`. A detailed description of all experiments along with additional visualizations and performance measurements can be found in appendix B.

**Poisson's equation** Poisson's equation, $\mathcal{P}(x) = \nabla^{-2}x$, plays an important role in electrostatics, Newtonian gravity, and fluid dynamics [4]. It has the property that local changes in $x$ can affect $\mathcal{P}(x)$ globally. Here we consider a two-dimensional system and train a U-net [48] to solve inverse problems (Eq. 1) on pseudo-randomly generated $y^*$. We compare SIP training to SGD with momentum, Adam, AdaHessian [55], Fourier neural operators (FNO) [34] and Hessian-free optimization (H-free) [39]. Fig. 6b shows the learning curves. The training with SGD, Adam and AdaHessian drastically slows within the first 300 iterations. FNO and H-free both improve upon this behavior, reaching twice the accuracy before slowing. For SIP, we construct scale-invariant $\Delta x$ based on the analytic inverse of Poisson's equation and use Adam to compute $\Delta\theta$. The curve closely resembles an exponential curve, which indicates linear convergence, the ideal case for first-order methods optimizing an $L_2$ objective. During all of training, the SIP variant converges exponentially faster than the traditional optimizers, its relative performance difference compared to Adam continually increasing from a factor of 3 at iteration 60 to a full order of magnitude after 5k iterations. This difference can be seen in the inferred solutions (Fig. 6a) which are noticeably more detailed.

**Heat equation** Next, we consider a system with fundamentally non-invertible dynamics. The heat equation, $\frac{\partial u}{\partial t} = \nu \cdot \nabla^2 u$, models heat flow in solids but also plays a part in many diffusive systems [16]. It gradually destroys information as the temperature equilibrium is approached [26], causing $\nabla\mathcal{P}$ to become near-singular. Inspired by heat conduction in microprocessors, we generate

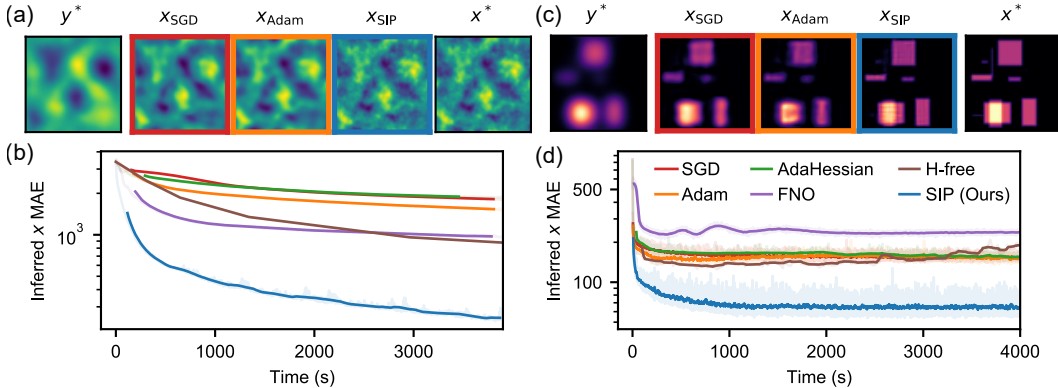

Figure 6: Poisson's equation (left) and the heat equation (right). (a,c) Example from the data set: observed distribution ($y^*$), inferred solutions, and ground truth solution ($x^*$). (b,d) Learning curves, running average over 64 mini-batches (except for H-free).

examples $x_{\mathrm{GT}}$ by randomly scattering four to ten heat generating rectangular regions on a plane and simulating the heat profile $y^* = \mathcal{P}(x_{\mathrm{GT}})$ as observed from outside a heat-conducting casing. The learning curves for the corresponding inverse problem are shown in Fig. 6d.

When training with SGD, Adam or AdaHessian, we observe that the distance to the solution starts rapidly decreasing before decelerating between iterations 30 and 40 to a slow but mostly stable convergence. The sudden deceleration is rooted in the adjoint problem, which is also a diffusion problem. Backpropagation through $\mathcal{P}$ removes detail from the gradients, which makes it hard for first-order methods to recover the solution. H-free initially finds better solutions but then stagnates with the solution quality slowly deteriorating. FNO performs poorly on this task, likely due to the sharp edges in $x^*$.

The information loss in $\mathcal{P}$ prevents direct numerical inversion of the gradient or Hessian. Instead, we add a dampening term to derive a numerically stable scale-invariant solver which we use for SIP training. Unlike SGD and Adam, the convergence of SIP training does not decelerate as early as the other methods, resulting in an exponentially faster convergence. At iteration 100, the predictions are around 34% more accurate compared to Adam, and the difference increases to 130% after 10k iterations, making the reconstructions noticeably sharper than with traditional training methods (Fig. 6c). To test the dependence of SIP on hyperparameters like batch size or learning rate, we perform this experiment with a range of values (see appendix B.3). Our results indicate that SIP training and Adam are impacted the same way by non-optimal hyperparameter configurations.

**Navier-Stokes equations** Fluids and turbulence are among the most challenging and least understood areas of physics due to their highly nonlinear behavior and chaotic nature [21]. We consider a two-dimensional system governed by the incompressible Navier-Stokes equations: $\frac{\partial v}{\partial t} = \nu \nabla^2 v - v \cdot \nabla v - \nabla p$, $\nabla \cdot v = 0$, where $p$ denotes pressure and $\nu$ the viscosity. At $t = 0$, a region of the fluid is randomly marked with a massless colorant $m_0$ that passively moves with the fluid, $\frac{\partial m}{\partial t} = -v \cdot \nabla m$. After time $t$, the marker is observed again to obtain $m_t$. The fluid motion is initialized as a superposition of linear motion, a large vortex and small-scale perturbations. An example observation pair $y^* = \{m_0, m_t\}$ is shown in Fig. 7a. The task is to find an initial fluid velocity $x \equiv v_0$ such that the fluid simulation $\mathcal{P}$ matches $m_t$ at time $t$. Since $\mathcal{P}$ is deterministic, $x$ encodes the complete fluid flow from 0 to $t$. We define the objective in frequency space with lower frequencies being weighted more strongly. This definition considers the match of the marker distribution on all scales, from the coarse global match to fine details, and is compatible with the definition in Eq. 1. We train a U-net [48] to solve these inverse problems; the learning curves are shown in Fig. 7b.

When training with Adam, the error decreases for the first 100 iterations while the network learns to infer velocities that lead to an approximate match. The error then proceeds to decline at a much lower rate, nearly coming to a standstill. This is caused by an overshoot in terms of vorticity, as visible in Fig. 7a right. While the resulting dynamics can roughly approximate the shape of the observed $m_t$,

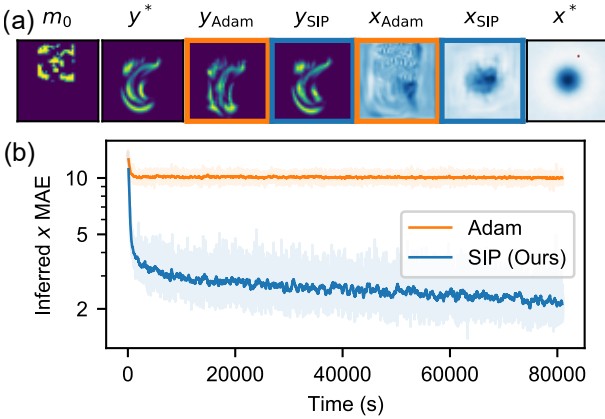

Figure 7: Incompressible fluid flow reconstruction. (a) Example from the data set: initial marker distribution ($m_0$); simulated marker distribution after time $t$ using ground-truth velocity ($y^*$) and network predictions ($y_\circ$); predicted initial velocities ($x_\circ$); ground truth velocity ($x^*$). (b) Learning curves, running average over 64 mini-batches.

they fail to match its detailed structure. Moving from this local optimum to the global optimum is very hard for the network as the distance in $x$ space is large and the gradients become very noisy due to the highly non-linear physics. A similar behavior can also be seen when optimizing single inverse problems with gradient descent where it takes more than 20k iterations for GD to converge on single problems.

For SIP training, we run the simulation in reverse, starting with $m_t$, to estimate the translation and vortex of $x$ in a scale-invariant manner. When used for network training, we observe vastly improved convergence behavior compared to first-order training. The error rapidly decreases during the first 200 iterations, at which point the inferred solutions are more accurate than pure Adam training by a factor of 2.3. The error then continues to improve at a slower but still exponentially faster rate than first-order training, reaching a relative accuracy advantage of 5x after 20k iterations. To match the network trained with pure Adam for 20k iterations, the SIP variant only requires 55 iterations. This improvement is possible because the inverse-physics solver and associated SIP updates do not suffer from the strongly-varying gradient magnitudes and directions, which drown the first-order signal in noise. Instead, the SIP updates behave much more smoothly, both in magnitude and direction.

Comparing the inferred solutions from the network to an iterative approach shows a large difference in inference time (table 1). To reach the same solution quality as the neural network prediction, $\mathcal{P}_{NS}^{-1}$ needs 7 iterations on average, which takes more than 10,000 times as long, and gradient descent (GD) does not reach the same quality even after 20k iterations. This difference is caused by $P_{NS}^{-1}$ having to run the full forward and backward simulation for each iteration. This cost is also required for each training iteration

Table 1: Time to reach equal solution quality in the fluid experiment, measured as MAE in $x$ space. The inference time is given per example in batch mode, followed by the number of iterations in parentheses.

| Method | Training time | Inference time |
|---|---|---|
| NN | 17.6 h (15.6 k) | 0.11 ms |
| $\mathcal{P}_{NS}^{-1}$ | n/a | 2.2 s (7) |
| GD | n/a | > 4h (20k) |

of the network but once converged, its inference is extremely fast, solving around 9000 problems per second in batch mode. For both iterative solver and network, we used a batch size of 64 and divide the total time by the batch size.

### 3.6 Limitations and Discussion

While SIP training manages to find vastly more accurate solutions for the examples above, there are some caveats to consider. First, an approximately scale-invariant physics solver is required. While in low-dimensional $x$ spaces Newton's method is a good candidate, high-dimensional spaces require another form of inversion. Some equations can locally be inverted analytically but for complex

problems, domain-specific knowledge may be required. However, this is a widely studied field and many specialized solvers have been developed [9].

Second, SIP uses traditional first-order optimizers to determine $\Delta\theta$. As discussed, these solvers behave poorly in ill-conditioned settings which can also affect SIP performance when the network outputs lie on different scales. Some recent works address this issue and have proposed network optimization based on inversion [39, 40, 20].

Third, while SIP training generally leads to more accurate solutions, measured in $x$ space, the same is not always true for the loss $L = \sum_i L_i$. SIP training weighs all examples equally, independent of the curvature $|\frac{\partial^2 L}{\partial x^2}|$ near a chosen solution. This can cause small errors in examples with large curvatures to dominate $L$. In these cases, or when the accuracy in $x$ space is not important, like in some control tasks, traditional training methods may perform better than SIP training.

## 4  Conclusions

We have introduced scale-invariant physics (SIP) training, a novel neural network training scheme for learning solutions to nonlinear inverse problems. SIP training leverages physics inversion to compute scale-invariant updates in the solution space. It provably converges assuming enough network updates $\Delta\theta$ are performed per solver evaluation and we have shown that it converges with a single $\Delta\theta$ update for a wide range of physics experiments. The scale-invariance allows it to find solutions exponentially faster than traditional learning methods for many physics problems while keeping the computational cost relatively low. While this work targets physical processes, SIP training could also be applied to other coupled nonlinear optimization problems, such as differentiable rendering or training invertible neural networks.

Scale-invariant optimizers, such as Newton's method, avoid many of the problems that plague deep learning at the moment. While their application to high-dimensional parameter spaces is currently limited, we hope that our method will help establish them as commonplace tools for training neural networks in the future.

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
