# OpenReview forum: "Scale-invariant Learning by Physics Inversion"
_NeurIPS.cc/2022/Conference — NeurIPS 2022 Accept_

### Official Review · Reviewer_3Np3 · 2022-07-11

**Rating:** 6
**Confidence:** 3
**Soundness:** 2 fair
**Presentation:** 2 fair
**Contribution:** 2 fair

**Summary:**

Inverse problems that involve physical systems usually have strongly varying gradients. This may lead first-order optimizers like SGD or Adam to compute suboptimal optimization directions. To handling this issue, the paper embeds a scale-invariant inverse problem solver into the gradient-descent-based learning pipeline. More specifically, the paper reformulates the objective function in the observed outcome space to that in the physical parameter space. It needs to compute the inverse of the physical process for each training step. Some theoretical and experimental results are provided.

**Questions:**

After reading the response and the comments from other reviewers, most of my concerns are addressed. So I increase my score to 6. My remaining concern is the approximate derivation in subsection 3.1. If there are no approximations, the performance is worse or better?
==================================================
There are some comments and suggestions.
1.	Highly related methods are missing for comparisons. There are many optimization methods to avoid compute the inverse of Hessian matrix. For example, the Hessian-free method. James Martens. Deep learning via Hessian-free optimization. ICML, 2010. It is better to include these methods for comparisons.
2.	The paper assumes that the physical process P has a unique inverse and is sufficiently well-behaved. It is better to give plentiful examples.
3.	At lines 134-147, the paper approximates P^{-1} with its first update. It needs more elaborations on this. What about iteratively solve P^{-1} precisely?
4.	At line 249, x has a square.


**Limitations:**

The paper did not have the limitations and potential negative societal impact.

**Strengths And Weaknesses:**

Strengths:

The paper presents a new approach to obtain scale-invariant updates of physical parameters, i.e., when rescaling the physical parameters, the optimization will behave the same way. Namely, it uses the objective function in the physical parameter space instead of the observed outcome space. Some acceleration tricks and convergence analyses are also presented. It is a useful complement to scale-invariant optimizers, e.g., Newton’s method, for high-dimensional parameter spaces.

Weakness:

See the comments for the "Questions” below.

---

> ### Author Response · Authors · 2022-07-31
> **Response to Reviewer 3Np3**
>
> > Highly related methods are missing for comparisons. There are many optimization methods to avoid computing the inverse of the Hessian matrix. For example, the Hessian-free method. James Martens. Deep learning via Hessian-free optimization. ICML, 2010.
>
> Our evaluation already contains comparisons to more modern techniques, such as AdaHessian (a successor of Martens 2010), RMSprop, Adam and many others. In our revised PDF, we have added comparisons to the original Hessian-free method from 2010 (Fig. 6). This method initially performs better than traditional optimizers on the heat equation experiment but quickly stagnates with the solution quality even decreasing after the first couple of iterations. On the Poisson problem, the Hessian-free optimizer takes up to half an hour to compute a single update, performing up to 250 CG iterations. Despite this, the Hessian-free optimizer performs better than the traditional methods like SGD, Adam and also AdaHessian but cannot match the accuracy of the SIP-trained network.
>
> > The paper assumes that the physical process P has a unique inverse and is sufficiently well-behaved.
>
> We do not make the assumption that P has a unique inverse. In fact, we allow the cases of multiple inverses or no perfect inverse at all. The goal of the optimization is to decrease the squared error ||P(x) - y||² (Eq. 1). When we talk about a solution x*, this need not be a perfect inverse; instead, we refer to a minimum of the above objective. In our derivation, we first derive an intermediate procedure that is limited to unimodal settings but our final method has no such limitations. The physics inversion can be performed locally around the current estimate, similar to how Newton’s method performs an inversion and is not limited to unimodal problems.
>
> > At lines 134-147, the paper approximates P^{-1} with its first update. It needs more elaborations on this. What about iteratively solving P^{-1} precisely?
>
> In the derivation, we simply expand the iterations of the iterative solver as ∑ₙ Δxₙ, as introduced directly below Eq. 1. Then we label the first term in that series U. Actually, we only require a function U which can be thought of as a vector field that always points towards lower loss values (definition in eqs. 11, 12). However, iterating U, i.e. U(U(...(x))), trivially defines an iterative solver that is guaranteed to converge. Having access to an iterative solver or U is therefore more or less equivalent. In our manuscript we start from the iterative solver viewpoint since that is more familiar to readers.
>
> > The paper did not have the limitations and potential negative societal impact.
>
> We explicitly discuss limitations in section “3.6 Limitations and Discussion”. As our work targets general inverse solvers for physics problems, we believe the paper has no negative societal impact, as stated in the checklist.

---

### Official Review · Reviewer_ukwX · 2022-07-11

**Rating:** 6
**Confidence:** 3
**Soundness:** 4 excellent
**Presentation:** 4 excellent
**Contribution:** 2 fair

**Summary:**

This work proposes scale-invariant learning for solving inverse problems. The state of the art machine learning techniques used to infer solutions to inverse problems rely on first order optimizers such as SGD or Adam. The highly non-linear behavior in physical processes result in suboptimal optimization direction. This work tackles this issue by proposing a 2 stage process which combines higher order optimization methods with Deep Learning. Specifically, their method uses the solutions from a scale invariant physics solver as targets for the neural network during training. Finally, they demonstrate that their method is much more effective than the existing methods on numerous problems such as - poisson's equation, Heat equation, Navier-Stokes equation.

**Questions:**

The authors have only compared their methods only against optimizers such as SGD, Adam, AdaHessian which seems a bit 1-dimensional. Currently the story is that SIP is better than the very traditional training pipeline and to me, the contributions seem limited to some extent. I believe the authors can paint a stronger picture if they compared their method against other popular approaches such as - neural operators [1] and Physics Informed Neural Networks [2]. I'm happy to increase my scores if the authors could compare their methods against these baselines.


[1] Kovachki N, Li Z, Liu B, Azizzadenesheli K, Bhattacharya K, Stuart A, Anandkumar A. Neural operator: Learning maps between function spaces. arXiv preprint arXiv:2108.08481. 2021 Aug 19.

[2] Karniadakis GE, Kevrekidis IG, Lu L, Perdikaris P, Wang S, Yang L. Physics-informed machine learning. Nature Reviews Physics. 2021 Jun;3(6):422-40.

**Ethics Review Area:**

["I don’t know"]

**Limitations:**

The authors have adequately addressed the limitations and potential negative societal impact.

**Strengths And Weaknesses:**

Strength:
* The 2-stage SIP pipeline Clearly outperforms the traditional training pipeline that uses gradient based optimizers to optimize the joint problem NN \odot P.
* This work has provided convergence proof for their algorithm.
* Empirically demonstrate the effectiveness of their work on problems such as - poisson's equation, Heat equation, Navier-Stokes equation.

Weakness:
The authors have only compared their methods only against optimizers such as SGD, Adam, AdaHessian which seems a bit 1-dimensional. Currently the story is that SIP is better than the very traditional training pipeline and to me, the contributions seem limited to some extent. I believe the authors can paint a stronger picture if they compared their method against other popular approaches such as - neural operators [1] and Physics Informed Neural Networks [2]. I'm happy to increase my scores if the authors could compare their methods against these baselines.


[1] Kovachki N, Li Z, Liu B, Azizzadenesheli K, Bhattacharya K, Stuart A, Anandkumar A. Neural operator: Learning maps between function spaces. arXiv preprint arXiv:2108.08481. 2021 Aug 19.

[2] Karniadakis GE, Kevrekidis IG, Lu L, Perdikaris P, Wang S, Yang L. Physics-informed machine learning. Nature Reviews Physics. 2021 Jun;3(6):422-40.

---

> ### Author Response · Authors · 2022-07-31
> **Response to Reviewer ukwX**
>
> > I believe the authors can paint a stronger picture if they compared their method against other popular approaches such as - neural operators [1] and Physics Informed Neural Networks [2].
>
> Thank you for the suggestions, we agree that both papers are very interesting approaches for inverse problems in physics. We have now added a direct comparison to a Fourier Neural Operator implementation for our Poisson and heat equation experiments. We found that a large model (width=32, modes=12 resulting in 1.2M parameters) works better than smaller variants. This model is significantly larger than our U-Net architecture which has only about 38k parameters. We trained the FNO using Adam and have found the optimal learning rate to be around 10^-4 for the heat equation and 0.003 for Poisson’s equation. With these settings, FNO works well for Poisson’s equation but has difficulties inferring accurate solutions for the heat experiment. Most likely, the nature of FNOs makes them amenable to the smooth structures of the Poisson solutions, but they have difficulties representing the sharp solutions of our heat equation problem. In both cases, SIP training with a regular network outperforms the FNOs. We have added the learning curves to Fig. 6 in the manuscript.
>
> PINNs are likewise a highly interesting approach, however, they do not directly compete with our method for two main reasons:
>
> 1. PINNs are typically employed to retrieve individual solutions, not learn distributions over data sets. When using them to solve the individual problems, inference is much slower since the network needs to be trained for each inferred solution. Iterative solvers seem like a better alternative in our setting.
> 2. PINNs work on continuous functions which is especially useful for sparse data. Our experiments are discretized on dense grids, negating many of the advantages of PINNs.
>
> We added a comparison to iterative solvers in our revision (Fig. 13). We believe this to be more appropriate but if you believe we should compare to PINNs as well, we’d be happy to perform and discuss the additional experiments.

---

### Official Review · Reviewer_8hA3 · 2022-07-14

**Rating:** 6
**Confidence:** 3
**Soundness:** 3 good
**Presentation:** 3 good
**Contribution:** 2 fair

**Summary:**

In this paper, the authors look at solving inverse problems with a neural network training scheme. They use a scale-invariant physics solver to compute the gradient update, and then train the neural network using this information. They show that this SIP training approach can converge faster to inverse problem solutions over optimizers such as Adam and SGD for various physical problems.

**Questions:**

- What is the advantage of the SIP solver vs. using the adjoint method as a way to back propagate through the weights?

- Can you clarify the problem setting better for the problems (such as Poisson and heat equations)? That is, exactly what the parameters being learned correspond to (i.e., by solving the inverse problem).

- How does the method compare to traditional numerical inverse solvers?

- Could the authors clarify what you mean by the network weights are not being altered?

**Limitations:**

Limitations are discussed.

**Strengths And Weaknesses:**

Strengths:

- Solving inverse problems is a long-standing problem in science and engineering, and the authors propose an interesting technique to integrate the physics solver with a neural network to address this challenge
- The method seems to converge more quickly compared to other optimization approaches, and can reconstruct solutions more accurately
- The paper is well-written for the most part

Weaknesses:
- The authors consider an infinite data setting, so it isn’t entirely clear what the generalization benefits of this method are
- The authors compare to mainly first-order optimizers and one second-order optimizer. It would be interesting to see how well the Newton’s method-based second-order optimizers do here, as are often used in solving numerical inverse problems (L-BFGS, etc.). There are also some promising results using other second-order optimizers, as well as exploring the use of starting with a first-order optimizer and then switching to a second-order optimizer.

---

> ### Author Response · Authors · 2022-07-31
> **Response to Reviewer 8hA3**
>
> > The authors consider an infinite data setting, so it isn’t entirely clear what the generalization benefits of this method are
>
> Our evaluation aims to compare the methods by their update steps. Therefore we set up our experiments to rule out as many other influences as possible, overfitting being one of them.
>
> To illustrate that our conclusions hold for training runs with finite data sets, we have performed multiple additional tests with finite data sets. There is no unexpected deviation between SIP training and traditional methods for reasonably-sized data sets and batch sizes. Both exhibit overfitting under the same circumstances, and the SIP training consistently outperforms Adam-based training across the varying dataset sizes. We have added a new figure, Fig. 12 in our revision, which shows the training and test curves depending on data set size and batch size. The runs of Fig. 12 show that there is no overfitting for the cases we discuss in the main text (batch size 128), and moderate data set sizes of 512 suffice to obtain representative results for the head equation case.
>
> > The authors compare to mainly first-order optimizers and one second-order optimizer. It would be interesting to see how well Newton's method-based second-order optimizers do here, as they are often used in solving numerical inverse problems (L-BFGS, etc.).
>
> An inherent problem with classical second-order methods like L-BFGS is that they do not work well with mini-batches since they estimate the Hessian by accumulating multiple updates. However, we have run gradient descent and L-BFGS-B on single examples, e.g. for the heat equation in Figs. 10, 13 in the appendix of the revised PDF. While our domain-specific iterative optimizers perform very well, standard solvers take a large number of iterations to match the network prediction. L-BFGS-B matches the Adam network after 500 iterations but slows down significantly before reaching the SIP network prediction. We believe that, given an unlimited optimization budget, both iterative solvers will eventually reach the accuracy of Adam and SIP, but will take many orders of magnitude longer.
>
> When it comes to training with mini-batches, AdaHessian is a more recent method that implements second-order optimization for neural networks. It can be thought of as an approximation to Newton’s method for batched optimization, which is why we chose it as a baseline. However, it performs poorly on the ill-conditioned problems we consider (Fig. 6), barely matching SGD measured against wall-clock time.
>
> > There are also some promising results using other second-order optimizers, as well as exploring the use of starting with a first-order optimizer and then switching to a second-order optimizer.
>
> AdaHessian is a prime example of a second-order optimizer for these problems, see question above.
>
> Starting with a first-order optimizer and switching to a second-order optimizer when approaching an optimum has the advantage of fast initial update steps combined with the quadratic convergence speed of second-order optimizers when the loss landscape approximately equals a quadratic form. This idea could be combined with SIP training in a number of ways, e.g. by pre-training the network using just Adam or by employing a second-order optimizer for the network weights later on. However, our SIP updates are not much more expensive to compute than first-order updates, so we expect the impact on our experiments to be low.
>
> > What is the advantage of the SIP solver vs. using the adjoint method as a way to back propagate through the weights
>
> We use SIP updates only for the physics solver to compute Δx. From there, we employ Adam to compute the weight updates Δθ. Adam uses backpropagation through the network, which is a special case of the adjoint method. We chose to keep the traditional network update with Adam because the ill-conditioning is typically caused by the physics. Neural networks can be designed and initialized to be well-behaved. If this does not answer the question, we’d be happy to clarify this aspect further.
>
> > Can you clarify the problem setting better for the problems (such as Poisson and heat equations)? That is, exactly what the parameters being learned correspond to (i.e., by solving the inverse problem).
>
> In both Poisson and heat equation, the physics simulation takes a scalar 64x64 grid, x, as input and computes a grid of the same dimensions, y, as output. The inverse problem consists of finding all 4096 parameters in x to satisfy the equation for a given y. When solving the inverse problem using a neural network, the network outputs the grid x.
>
> We answer the remaining questions in the next comment.

---

> > ### Author Response · Authors · 2022-07-31
> > **Continuation of the Response to Reviewer 8hA3**
> >
> > > How does the method compare to traditional numerical inverse solvers?
> >
> > In theory, traditional iterative solvers can surpass the network predictions in accuracy but at the cost of much higher run-time. Table 1 shows this for our Navier-Stokes experiment. Our domain-specific scale-invariant solver reaches the same accuracy as the network after 7 iterations while a standard optimizer will take thousands of iterations and not find the best solution.
> >
> > We also tested the gradient descent and L-BFGS-B optimizers on the heat equation. Figure 13 in the revised PDF shows the optimization curves. As expected, L-BFGS-B performs better than gradient descent, matching the prediction from the neural network trained with Adam after 500 iterations. However, both optimizers fail to reach the accuracy of the near-instantaneous (64 ms) predictions of the SIP network within a reasonable time frame. Running L-BFGS for 1000 iterations took 102 seconds. The results are visualized in Fig 10 top.
> >
> > > Could the authors clarify what you mean by the network weights are not being altered?
> >
> > We assume you are referring to the sentence “Our approach does not alter the update rule for the network weights themselves”. We have clarified this statement in our revision. Our aim was to state that we use the adjoint method with a first-order optimizer like Adam to compute the update for the network weights. SIP updates only replace the physics gradient, and in this way can be coupled with all first-order methods (Adam, SGD, AdaGrad, etc.) for training neural networks.

---

### Author Response · Authors · 2022-07-31
**General Response to the Reviews**

We would like to thank the reviewers for their valuable feedback!

We have uploaded a revised version of our manuscript PDF, adding additional figures to the appendix. Changed sections are highlighted in blue. We answer the questions and concerns of the reviewers in the individual responses below.

---

### Meta-Review · Area_Chair_qfFZ · 2022-08-26

**Recommendation:** Accept
**Confidence:** Certain

**Metareview:**

Reviewers agreed that solving inverse problems is a long-standing problem in science and engineering, and the authors propose an interesting technique to integrate the physics solver with a neural network to address this challenge.

**Award:**

No

---

### Decision · Program_Chairs · 2022-09-14

Accept